# What slows the progress of health systems strengthening at subnational level? A political economy analysis of three districts in Uganda

Justine Namakula[1]*, Xavier Nsabagasani[1], Ligia Paina[2], Abigail Neel[2], Chimwemwe Msukwa[3], Daniela C. Rodriguez[2], Freddie Ssengooba[1]

1 Department of Health Policy Planning and Management, School of Public Health, Makerere University, Kampala, Uganda, 2 School of Public Health Johns, Hopkins University Bloomberg, Baltimore, Maryland, United States of America, 3 UNICEF Country Office, Kampala, Uganda

* jhasteen@yahoo.com

## Abstract

There is increasing recognition that without stronger health systems, efforts to improve global health and Universal Health Coverage cannot be achieved. Over the last three decades, initiatives to strengthen health systems in low-income countries have attracted huge investments in the context of achieving the Millennium Development Goals, the Sustainable Development Goals, as well as Universal Health Coverage. Yet, health system inadequacies persist, especially at the subnational level. Our paper presents a political economy analysis featuring a three-district case study in Uganda, where district-based health systems strengthening initiatives were implemented. The study sought to understand why health systems at the subnational level are failing to improve despite marked investments. This problem-based political economy analysis draws from a document review and key informant interviews [N = 49] at the central and district levels with government actors, development partners, and civil society in three purposively selected districts. Available financial data extraction and analysis were used to complement qualitative data. We found that challenges in strengthening district health systems were numerous. Themes related to financing and planning broadly interacted to curtail progress on strengthening subnational-level health systems. Specific challenges included inadequate financing, mismatch of resources and targets, convoluted financial flows, as well as unwieldy bureaucratic processes. Sticky issues related to the planning process included variations in planning cycles, conflicting interests among actors, insufficient community engagement, limited decision space, and distorted accounting mechanisms. In conclusion, the political economy analysis lens was a useful tool that enabled an understanding of the dynamics of decision-making and resource allocation within district health systems, as well as the performance in terms of implementation of the district work and plans with existing resources. Whereas the district health teams play

**Data availability statement:** All data relating to this study is available at Makerere University School of Public Health can be availed in conformity to institutional data governance policy. Data requests should be forwarded to sphrecadmin@musph.ac.ug and sengooba@musph.ac.ug.

**Funding:** This study was primarily funded by the United Nations Children's Fund (UNICEF) with funding from the Bill & Melinda Gates Foundation (contract #43297444) to DR and LP at Johns Hopkins University, Bloomberg. Johns Hopkins University then sub-granted FS, JN and XN at Makerere University School of Public Health to work collaboratively. The views expressed in this manuscript do not necessarily reflect views of UNICEF and Bill and Melinda Gates Foundations. The findings do not represent the Official position of UNICEF Uganda. The broader funders at UNICEF had no role in study design, data collection, analysis, decision to publish or preparation of the manuscript.

**Competing interests:** The authors have declared that no competing interests exist.

a big role in service program implementation, the context in which they work needs to be improved in terms of sufficient resources, setting realistic targets, widening the decision space and capacity necessary to engage with other various stakeholders and effectively harmonize the implementation of the programs. Despite playing a crucial role in compensating for local shortfalls in resources, donor resources and engagements should not happen at the cost of the subnational voice in priority setting and decision-making.

## Introduction

There is widespread recognition that efforts to improve global health and to achieve Universal Health Coverage (UHC) cannot be achieved without stronger health systems [1]. As a result, over the last three decades, Health Systems Strengthening (HSS) in low-income countries has attracted many investments in the context of achieving the Millennium Development Goals (MDG), the Sustainable Development Goals (SDG), as well as UHC reforms [2]. Yet, the goal of UHC – from "health for all by 2000" to "UHC by 2030" – has remained elusive for the last 30 years, as health system inadequacy, especially at the subnational level, has persisted. Important contributors to this issue are the political will to sustain a UHC agenda, as well as government underinvestment in health systems, high dependency on development assistance, a lack of fiscal space, insufficient attention to primary health care, and under-developed health financing systems, which maintain health systems' reliance on out-of-pocket payments [3,4]. Although general budget support has long been recognized as a helpful means to support national priorities, including towards UHC, not all donors have re-aligned there investments as such and country governments also continue to face challenges with public financial management [5].

In Uganda, while consistent progress has been made toward achieving UHC [6], and multi-donor general budget support has enabled the government to implement several pro-poor policies [5], such as the removal of user fees for certain groups, there has been significant variation in subnational performance (i.e., gaps in effective coverage of key interventions, high out-of-pocket costs) [7–10]. In this context, both national and global actors are concerned about sub-optimal returns on HSS investments in terms of improving health system performance, resilience, and the sustainability of the capacity thereof [11], particularly as subnational levels of the health system are hypothesized as being critical to advancing progress toward UHC [12,13].

UHC strives for all people to access quality health services "when and where they need them, without financial hardship [14] UHC is often operationalized through health insurance coverage and as the basket of free-of-charge services [15], but the HSS agenda underscores the role of UHC in ensuring there is benefit to all people in a defined functional zone [11], giving priority to the district and subnational efforts for health systems strengthening [16,17]. To achieve UHC through Primary Health Care (PHC)-oriented health reforms, WHO established the concept of district health systems aimed at creating a functional zone for planning and implementation of PHC

services at the subnational level [18,19], enabled within the decentralization framework. This stimulated devolution and decentralization efforts within the health system and renewed the need for decision-makers to understand the interplay between national and subnational levels, the politics of resource allocation, and the implications of health systems reform efforts on the services that the population receives.

Despite the breadth of lessons learned about health systems reforms and HSS, a lot remains to be uncovered. Progress has been made in analyzing health systems functions and performance through the core building blocks, health system process elements, and priority outcomes [18]. More recently, the literature has focused increasingly on the interactive nature of the health systems building blocks to generate new understandings about interactivity, dynamism, and resilience [4,20–22]. Although acknowledged as important, less progress has been registered about the multiplicity of actors interacting to oversee and deliver services, their power and interests, and how these can be coordinated for common system goals [23,24]. Yet, these are major challenges of multi-actor capabilities, governance, and institutional arrangements that underpin health systems strengthening agendas.

In Uganda, decentralization was piloted in a few districts in 1993 as one of the hallmarks of reforms to improve service delivery across all government programs [25]. Later, in 1997, the national resistance movement passed a Local Government Act 1997 [26], later amended in 2001, 2005, and 2006 [25]. The Local Government Act of 1997 provided a framework to rationalize the hierarchy of decentralised administrative units from village to district level [25]. Local governments (LGs) hold the mandate for planning, spending, and administration of programs for a defined district or municipality [26]. Like in many other countries, the LGs in Uganda have healthcare and public health mandates and form the basic unit of a health system at the subnational level [27]. Over the years, district health systems have become an essential locus for improving the health system to deliver PHC [28]. Indeed, international organizations such as the WHO and UNICEF have underscored a renewed focus on PHC- as a vision and operational framework for achieving UHC and sustainable development goals [26]. As such, district health systems have attracted investments from international development partners, who have also sought to understand how the local decision-making environment may affect the implementation of health initiatives. Therefore, an examination of political economy factors at the subnational level, (i.e., factors related to the interaction of political and economic processes, the distribution of power between groups, and the processes that create, sustain, and transform relationship within a system over time (UK-DFID, 2009)), and their influence on the effectiveness of investments in sub-national systems, is timely.

This article is based on a three-district case studies about the political economy of subnational HSS in Uganda. The study sought to understand how the current political economy, particularly issues around health management, governance, and finance, have affected the development and management of district-level health systems. More specifically, our study set out to explore a) how the financing and planning ecosystems function to support HSS at the subnational level and related challenges, and b) what are the plausible solutions to address these HSS plans and priorities at the subnational level?

## Materials and methods

This paper presents data from Uganda, drawn from a larger study of the UNICEF-supported District Health Systems Strengthening Initiative (DHSSi) in Kenya, Malawi, Tanzania, and Uganda [29]. The goal of DHSSi was to improve subnational health management and governance as a critical component of HSS necessary to achieve UHC. DHSSi activities were focused on capacity building for evidence use in district planning, implementation, and performance management for district health management teams as well as supporting the scale-up and professionalization of health management [29]. During implementation, DHSSi identified several management gaps and challenges in the enabling environment that make good management practice difficult [30] and required a better understanding of the political economy at play at the subnational level. In 2019, UNICEF requested a political economy analysis of DHSSi in Kenya, Malawi, and Uganda to explore these issues. The study was conducted through a collaboration between Makerere University School of Public

Health and Johns Hopkins University. Findings from the sister studies in Kenya and Malawi, as well as a cross-country synthesis of the political economy analysis findings, are available elsewhere [31,32].

## Political economy analysis (PEA) framework

We used a problem-driven Political Economy Analytical (PEA) tool- Fig 1, useful for understanding the contextual factors, structural factors, and related rules of the game, actors, relationships and roles, motivations and interests as well as change process and coping mechanisms. (UK-DFID, 2009; Harris, 2013; Fritz, 2014).

The PEA framework focuses on a specific problem or policy to better understand a challenging issue and the institutional dynamics contributing to the problem, including the broader actors and systems factors that facilitate or hinder change [33–35].

The framework has 3 main rows, with the starting point being the top row which relates to problem identification and breaking down the characteristics/ manifestations of the problem. The problem, as defined in this study relates to 'The enabling environment hinders the performance and management of the health systems at the local level' which hinders the successful implementation of health services and health systems strengthening activities, and, as a result, leads to a series of poor performance outcomes at sub-national levels. The problem is manifested through outcomes such as underspent budgets indicated at the top right corner of the framework. The second phase is to conduct a detailed two-step diagnosis, that is structural and agency diagnosis, to understand why the problem persists. The structural diagnosis focuses factors including historical and demographic factors as well as norms, rules, and procedures. The agency

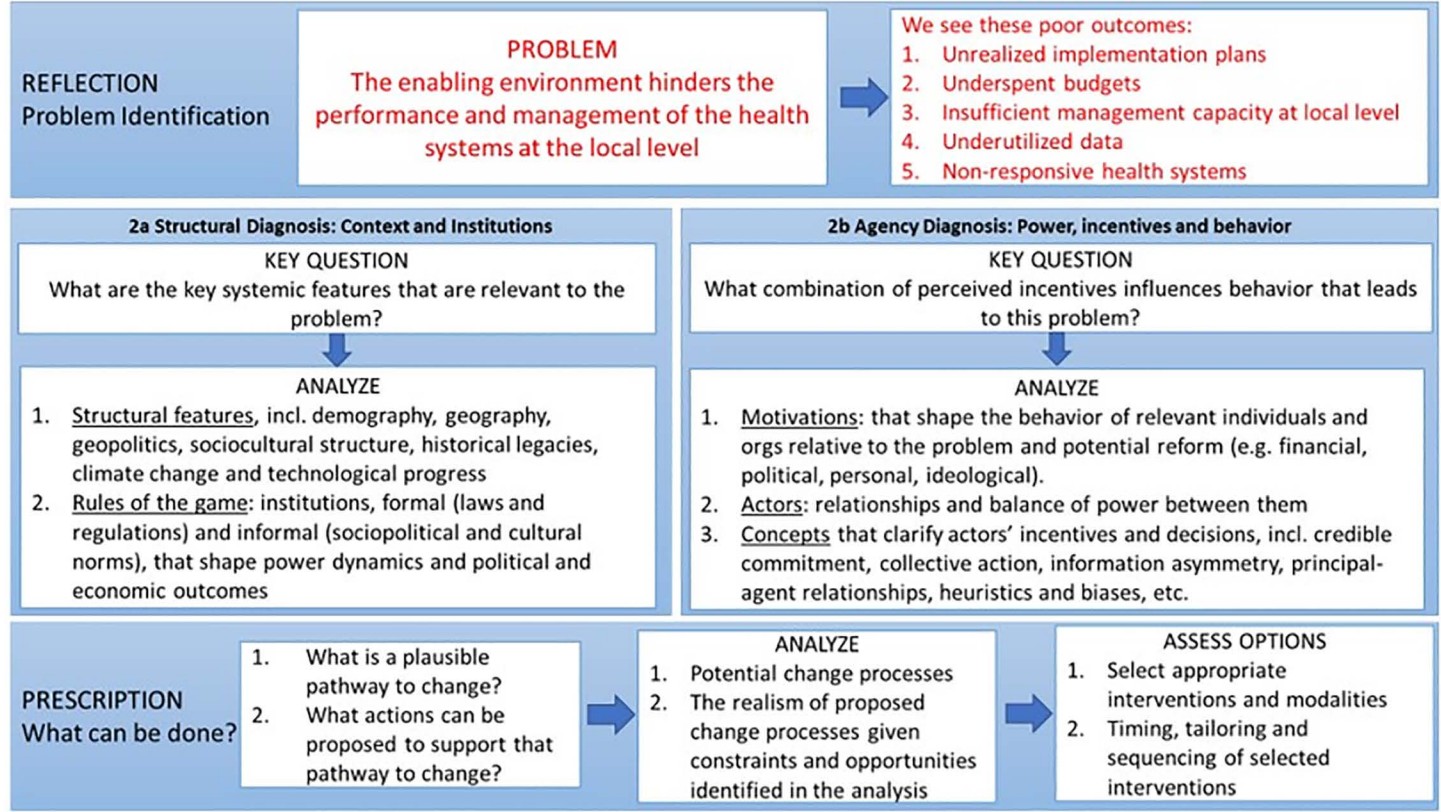

**Fig 1. Analytical pathway for the problem-driven.** PEA, adapted from Siddiqi et al. (2009).

diagnosis addresses the key actors involved, roles, power dynamics as well as interests and motivations driving their behaviors. After the problem diagnosis, political economic analysts then identify plausible pathways for change and coping mechanisms in the short, medium, and long term (lower row of the framework).

**Study design.** The study team implemented a three-district case study design that employed predominantly qualitative data collection methods including document review and key interviews (KIIs). The qualitative data was complemented with some secondary data extraction for financial information [30]. Case selection criteria: Three districts were purposively selected in collaboration with UNICEF-Uganda office out of the ten total districts in which the DHSSi project was implemented from 2019–2022 [36]. The study districts were Iganga, Kiryandongo, and Isingiro [30]. The case selection criteria included geographical location, year of establishment, having participated in the UNICEF DHISS project, and performance on the National League table in terms of performance, we tried to select a mix of good performance and bad performance. For example, Kiryandongo was a good performer, while Iganga had average performance and Isingiro was the least performer. We also considered special contextual factors such as being host to refugee settlements (Isingiro and Kiryandongo) and poverty.

## Data collection

Data collection methods included key informant interviews (KIIs), document review, secondary data extraction and two stakeholder dissemination and validation meetings. These are summarized in Table 1 and explained further below.

**Key informant interviews.** We conducted 49 key informant interviews (KIIs) between November 2020 and April 2021 with purposively selected respondents. Respondents included representatives of local and central government ministries, funding agencies, and implementing partners.

The interviews were conducted using an unstructured thematic guide derived from the problem-driven Political Economy Analysis (PEA) conceptual framework. At subnational level, the guide focused on experiences of the planning and budgeting process, both of which were integral to DHSSI activities, identification of key stakeholders/actors in the planning process, their roles, power/influence, interests and relations between actors and the politics of resource allocation and coordination between actors. Furthermore, the interview focused on the timing and feasibility of the planning process as experienced by sub-national level actors, rules of the game, challenges and coping strategies/ innovations to game the rules. Other issues probed during the interviews related to the monitoring and accountability processes and how these affect planning, budgeting and implementation of activities.

At national level, the interview guide focused on oversight role of the Ministry of Health, explanations about the national resource allocation priorities and late disbursement challenges as well as developments related to community engagement All interviews were conducted in English. District-level respondents participated in interviews in person. Either

**Table 1. Summary of the categories of participants and number of interviews across districts.**

|  | KIIs | Participant categories |
|---|---|---|
| District 1 | 14 | District Health Officers, District planners, Biostatisticians, Chief Administrative Officers, Local Council (LC5) Chairpersons, Village Health Team members, Health Unit Management committees, Health Inspectors, Health Sub-district In-charges |
| District 2 | 14 | |
| District 3 | 13 | |
| Central Government | 3 | Ministry of Health |
| Implementing partners | 5 | |
| Total number of interviews | **49** | |
| Gender | | |
| Female | 9 | |
| Male | 40 | |

national level respondents participated in person or remotely, as was feasible during the data collection period, which coincided with several COVID-19 pandemic-related meetings and movement restrictions. Interviews were audio recorded with permission from respondents.

**Document review.** For the document review, over 40 documents on Health systems strengthening (HSS), guidelines for policy and planning, as well as financial management were pragmatically identified via google search, Ministry of Health website and Library and recommendation by key informants. The type of documents reviewed included grey literature and reports on the implementation and evaluation of programs, government reports and policies, and some peer-reviewed articles.

**Secondary data extraction.** Financial data was obtained from the district administrations, on request and extracted using a template generated by the research team.

**Stakeholder validation meetings.** We organised two virtual stakeholder dissemination and validation meetings. Participants included stakeholders of the health sector budget- working group, including Ministry of Health (MOH) officials and other development partners, as well as selected district leaders. As part of this process, our team also led discussions around potential solutions to the challenges identified.

## Data analysis

**Document analysis.** Data from the document review was analyzed using content analysis. To organize the data, we used a simple assessment matrix with categories such as type of document, date, information/ content that relates to components (e.g., resource allocation, prioritisation, procedures and processes) and implications. Emerging themes from the document review were integrated into the findings of the KIIs and so were findings from the data extraction.

**Analysis of KIIs.** The Framework Analysis method was to guide the analysis and this was similar to all the participating country teams [33]. Broadly, we followed the 7 steps outlined in Fig 2. Audio recordings were transcribed with verbatim, and the transcripts were reviewed by the country team and their JHSPH backstops.

An abstraction matrix was developed in MS Excel following the domains in the PEA analytical framework by Siddiqi et al 2009 [37], that is structural diagnosis, agency diagnosis, and change. Each matrix tab was focused on specific themes and sub-themes. As transcripts were reviewed, short summaries were abstracted into the relevant cell in the tab. Each respondent was captured in one spreadsheet row.

The analytical framework was revised through two rounds of testing. In the first round of testing, the JHSPH central team tested the abstraction matrix using transcripts from all three countries. The team reviewed the matrices for consistency, discussed reliability among abstractors, and modified the matrix and its related themes and

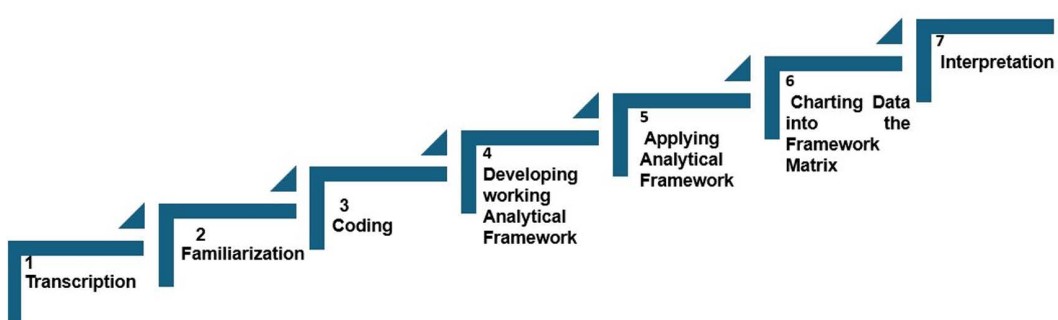

**Fig 2. Steps in framework analysis.**

sub-themes accordingly. Based on this initial round, the JHSPH team developed a pre-recorded training session for country teams – orienting them to the framework and the overall data analysis process. The training was shared online for asynchronous viewing then reviewed during a cross-country call, allowing for discussion and clarifications.

In the second round of testing, each country team and their JHSPH backstops tested the abstraction matrix with a transcript and discussed emerging questions and made a final round of minor adjustments to the matrix. After the matrix was finalized, each country team proceeded with the data abstraction. The final abstraction which was used by all country teams matrix is uploaded as S1 Data.

Synthesis of data was done by theme and sub-theme across respondents. In the matrix, teams added one synthesis summary per column, so that each sub-theme can have multiple threads of findings. Country teams held internal meetings to discuss emerging themes and to interpret and explain the data, focused on the Structural and Agency Diagnoses and Pathways for Change inherent to problem-driven PEA. Initial recommendations for overcoming environmental and governance barriers to effective health management were developed through data analysis and then revised and elaborated via stakeholder consultations. Further details about the data analysis approach can be found in the cross-country paper resulting from this project [33].

Financial data analysis: As the completeness of the budgetary and expenditure data varied from district to district, the analysis focused on financial trends for the budgets and expenditures for three fiscal years (2018–2019; 2019–2020 and 2020 – 2021). Comparisons were made for budgets allocations for government and donor funding. Data on health spending at the sub-national level was fragmented across many sources of funds.

### Ethics approval

The study received an exemption and approval from the Johns Hopkins School of Public Health (JHSPH) ethics board (IRB.no. 12934). The protocol was submitted and cleared by the Higher Degrees, Research and Ethics Committee (HDREC) of Makerere University School of Public Health (Protocol No. 890), and Uganda National Council for Science and Technology (UNCST) for approval (SS664ES).

### Ethical statement

Formal written consent was obtained from all study participants at both national and subnational level. Given the political nature of the study, participants were assured of anonymity. The study team also decided to give codes according to the category of organization level that was being represented by a particular participant. For instance, terms such as "district respondent" and "IP respondent" were used. Within the text for findings, the districts were blindly given numbers 1, 2, 3, for anonymity purposes. The district name to which the number was assigned is only known to the study team. All data was stored on a password-protected computer, accessed only by the research team members.

Furthermore, the study was conducted during the COVID-19 pandemic. As a result, the Ministry of Health provided other extra standard operational procedures such as social distancing, use of masks, and hand washing/sanitization, which were also strictly observed by the study team. While the study team had planned a mix of virtual and in-person interviews, respondents at the district level preferred the latter.

### Results

In our findings, we touch upon associated constructs from the structural and agency diagnoses of our adapted political economy analysis framework to explain root causes of the problem. Overall, we found that financing challenges and bureaucratic procedures in the planning process and actors' interests, influence, and relationships interact to curtail progress on HSS at the subnational level.

## Structural diagnosis and related findings

Our structural diagnosis revealed slow progress was mainly reflected in the financing systems for decentralised service delivery and related rules and bureaucracies. The sticky issues around financing related to insufficient amount of funds disbursed, delays in disbursement, insufficient locally generated revenue, resource allocation priorities, mismatch between resources and targets, and duplicative bureaucracies for implementation and related perceptions among actors. We will elaborate further on each of these sub subjects in the next sections.

**Insufficient financial resources at the sub-national level reinforce dependence on funding streams associated with priorities set at higher levels in the system.** Our analysis of both KII and the available financing data confirmed the existence of grossly insufficient financial resources allocated from the Central Government to LGs, relative to the population size and expected obligations in healthcare service delivery. Several KII respondents suggested inadequate financial and non-financial resources at the subnational level as the most pervasive problem to all HSS efforts. Findings from the document review pointed to insufficient national revenue to the health sector as well as inflation as primary reasons for consistently low resource allocation to LGs [38]. Our financial analysis of available data on government allocations from financial years 2018/19, 2019/2020 and 2020/2021,(Fig 3), illustrated a negligible increase in funding to the study districts.

Furthermore, funding allocations across study districts did not appear to account for contextual differences across districts in terms of population size, location, year of establishment, and other demographic factors. The main source of funding from the Ministry of Finance and channeled through Ministries of Health and local government is the Conditional Primary Health Care (PHC) allocation, disbursed quarterly. PHC funding covers several activities linked to national level-determined priorities and is accompanied by guidelines and budget items, hence providing LGs with limited flexibility for reallocation at the district level.

Although the available financial data across all three districts points to stable resource allocations, most of our respondents reported a perceived reduction in financial allocations from the central government. Findings from the document review indicated the perceived reduction in financial flows stemmed from the loss of value resulting from (1) inflation, and (2) the rapid proliferation of districts and administrative units.

Further complicating resource availability, insufficient financial resources from the central government created high district dependence on funding streams that have pre-determined priorities, such as donor funding whose priorities are determined at the global or national level, without consideration of specific subnational problems, priorities, and needs.

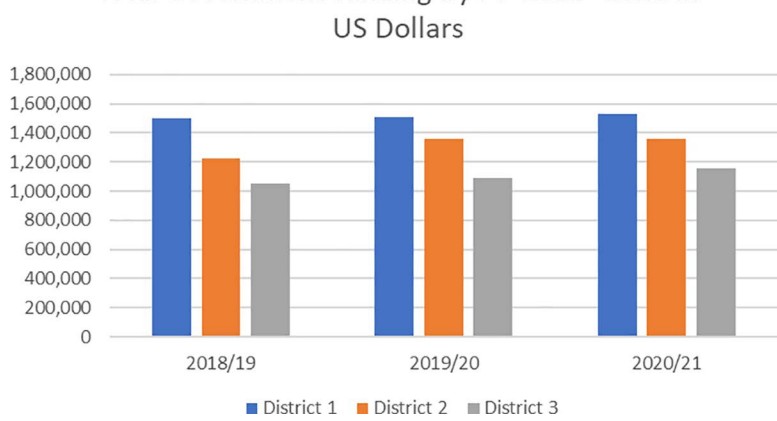

**Fig 3. Total government funding by financial year (2018-2021) in US Dollars.** Source: Financial data extracted from study districts.

According to the respondents' reflections, such externally determined funding conditions directly affected planning and the local decision space and were often at the expense of local actors' priorities. For example, our respondents reported concerns that resource allocation is biased towards greater investments in "software," aimed at improving information technology (IT), trainings, and behavioral change communication, data for planning, report cards and community engagements for accountability. In contrast, subnational respondents reported several major "hardware" investment priorities that are needed to strengthen subnational health systems, such as infrastructure, health workforce strengthening, building maternity homes, providing more nurses and doctors, building accommodation for health workers, and increasing the stocks for medicines. Most hardware priorities were focused on infrastructural development, which, except in very few instances, tended largely to be ignored by both the central government and implementing partners (IPs). In a few instances, there was a mention of implementing partners focusing more on renovations and refurbishments. Nevertheless, as stated by some respondents, such hardware investment needs perpetually remain on the list of "unfunded" priorities, due to a mismatch in priorities between donor-funded IPs and local priorities, and the system remains fragile.

"[…] [infrastructural developments] remain unfunded because there is always no money, and Partners do not want to fund such things…they [IPs] reached an understanding amongst themselves that they do not want to go through capital development." (District Respondent_ 07)

**Delays in disbursement of funds, leading to coping mechanisms and exacerbating dependency on donor funding.** Across all study districts, respondents reported significant delays in receiving available resources, significantly delaying the implementation of district work plans. While there were reports of improvements in the timeliness of disbursements in District 2 rather than district 1, coping strategies innovated by subnational managers from Districts 1 and 2 were needed to ensure the implementation of health activities. As highlighted in the quote below, these included asking health workers to "loan their time'," and district offices taking loans/debt from local service providers, including petrol stations and stationery shops, for the needed services and commodities. In both cases, subnational managers mentioned that, in extreme situations, they were forced to delay or cancel the implementation of some of the activities.

"And yet getting the money is not quick […] we go ahead and do the activity on debt. […] We do the accountability and pay later. So […] requesting the people [health workers] to do the activity […] as we are processing […] we go to fuel stations and borrow the fuel then we tell the people to do the work." (District respondent_07_district 2)

Respondents from District 3 did not mention similar coping strategies but rather reported they leveraged support from implementing partners.

Across all three districts, there were gaps in financial data, although district 3 was worse off as it lacked data for two financial years. In relation to financial data from donors compared to funds from government for specific service delivery areas, there were data gaps as well. The amount of donor funding was greater than government funding for most of the technical areas where data existed (Table 2).

**Insufficient and 'inequitable' local tax revenue distribution between health and other sectors.** The Local Government Act 1997 enables local governments to collect taxes as a reliable source of funding [26]. The local revenue generated would ideally provide a fallback option when delays occur with central funding. Besides, the local revenue is earmarked to fund areas such as salaries for the staff not on government payrolls. However, in addition to local tax revenues being limited, politics at the district level rendered it difficult for the health department to access such funds. For example, in one of the study districts, politicians at the local government level perceived the District Health Office to be "richer than other departments" because it had many externally funded implementing partners supporting activities. Therefore, monies were either siphoned by those in charge or "preserved for worse off departments," hence hindering the implementation of some operational activities for health.

Table 2. Comparative analysis of government vs donor funding by technical areas.

| Technical area | FY 2018/19 (USD) | | | | FY 2019/20 (USD) | | | |
|---|---|---|---|---|---|---|---|---|
| | District 1 | | District 2 | | District 1 | | District 2 | |
| | GOU | Donors | GOU | Donors | GOU | Donors | GOU | Donors |
| 1.Technical support supervision-DHT team | 1,444 | 2,222 | 9,611 | DNA | 333 | DNA | 11,791 | DNA |
| 2.Support supervision - Integrated (Political+DHT) | DNA | DNA | 7,922 | DNA | 1,111 | DNA | 10,005 | DNA |
| 3.Immunization activities | 18,056 | 66,667 | 6,111 | DNA | DNA | 166,667 | 6,111 | DNA |
| 4.Maternal newborn and child health activities | 889 | DNA | DNA | DNA | DNA | 222,222 | DNA | DNA |
| 5.Nutritional activities | DNA | DNA | DNA | DNA | DNA | DNA | DNA | DNA |
| 6.TB activities | 444 | 1,812 | DNA | DNA | 1,700 | DNA | DNA | DNA |
| 7.HIV activities | 4,444 | DNA | DNA | DNA | DNA | 27,778 | DNA | DNA |
| 8.HMIS (Data supervision) | 667 | 473 | 833 | DNA | 1,667 | DNA | 1,000 | DNA |

Note: Dollar equivalent at rate of 3600/=. DNA=Data not available, Source: financial data extraction and analysis by study team.

"[…] we are not able to give a good helping hand into the health sector, given the availability of some donors in the department than any other department. So, having very little PHC has again caused issues at the facilities because you may find a very bushy compound, facilities don't have fencing […] (District Respondent_05)

**Mismatch between insufficient resources and overly ambitious service delivery targets or expectations.** The mismatch between the service delivery targets and resources [financial and human resources] also emerged as a key issue. The respondents attributed this issue to an inadequacy of resources, coupled with overwhelming local community needs and expectations, unrealistic demands from the central government, and the general dilemma of cost-efficiency (i.e., desire to do so much using minimal resources). The study found that many frustrations about HSS interventions at the subnational level arose from overly ambitious expectations from IPs and the central government despite little resources made available to LGs. As reflected by district respondents' experiences in the case of the Measles-Rubella Campaign of 2020, expectations of targets were misaligned with operational costs and realities on the ground, with limited room for negotiation with central government about increment. As illustrated by the quote below, the district teams did not have the opportunity to negotiated with the Ministry of Health or provide revised data and coped by over stretching both the limited financial and human resources available to achieve outputs. It is not clear if the targets were met.

"For measles rubella, the Ministry of Health […] thought for us […] and they said that district Y, we are giving you this much and you are going to implement it this way. Of course, we wanted to tell them that it might not work. Let me give you an example. They calculated that we have 200 and something schools, but here the [District Education Officer] DEO had 938 schools [in the records]. Now, […] on the ground, we also found day care centres, which were operating without DEO's knowledge. So, we had to work within the available budget to meet all this demand. We had to arm-twist [staff] […]" (District Respondent_04).

Furthermore, concerning planning for human resources for health, one of the district managers acknowledged the mismatch of needs and targets as an apparent problem that hindered plans for recruitment and filling of staffing gaps, with district managers resorting to identifying ways of uncomfortably managing activities with minimal staff.

"[…] you cannot implement something for which you know there is no money. For example, if you have 400 health workers and they are taking 4 billion and [yet] you need 700 health workers [who] will take 10 billion you cannot just say I want 700 health workers […], because the resource envelope is small. So […] we have not got funding for all the needs because the needs are far more than the resources." (District Respondent_17)

## Agency diagnosis and related findings

The agency diagnosis revealed existence of exist various actors, with different power, influence conflicting interests and motivations among HSS actors at sub-national level. Notably, the relationships among actors were also not always positive. The complexity of actors and relationships further created more challenges including variations in planning cycles and distorted accountability mechanisms, shortcuts to community engagement. In the sections below, we attempt to enlist the different actors and challenges related to their power[powerlessness], perceived relationships with others and how this affects the functionality of the system.

**Many local government actors: conflicting interests and motivations.** The study found a proliferation of actors within the subnational level planning space, with varying interests and power, crowding the decision space. These actors are located at both the district and national levels, and prioritizing sectoral investments, often outside of health, that can maximize the votes they could obtain from the local community. At the subnational level, actors include district level managers, technical teams, extended district management teams, administrators, political leaders, implementing partners, and the community representatives, often with competing priorities, as highlighted by the quote below.

> "[…] our new Chief Administrative Officer (CAO) said that […] health must benefit at least by supporting one facility per year […]. Our LC V chairman, for him he was interested in education because he realized that building a classroom was much cheaper than building a health facility. […] build very many classrooms so that at the end of his term he can say that 'I have built these many classrooms' and then he gets his votes". (District Respondent_04)

Engagement of many actors in the local planning processes increases transactional costs and creates a congested space for decision-making. In addition, the influence of politicians at the subnational level during the planning process led to diversion of discussions and posed a potential effect on outcomes for prioritization in exchange for votes.

**Community [non] participation – challenges and shortcuts.** Community participation in planning and accountability is a crucial part of planning and budgeting process at sub-national level, with community needs contributing to the shaping of priorities at sub-national level. However, our agency diagnosis revealed that community engagement is limited, and that the community is largely silent and powerless. Furthermore, we found that the definition and operationalisation of community participation in planning vary in study districts and that community participation was perceived to be slow and very costly. Hence, there were many shortcuts to community participation. These included using routine health data to inform district plans, using data from previous plans, or pseudo representation of the community using one member. As explained by one of our respondents, even where facility management committees exist, they have little power in practice and community recommendations may be influenced by more powerful health system actors.

> [...]Sometimes we sit and think that these health facility management committees are [very] powerful in making things work but not much. […] they cannot go beyond recommending. Moreover, their 'recommendations' are probably crafted probably by the medical officer or the person in charge or whoever and it may not be really biting. (District Respondent _11)

As a result, the communities lost trust in the planning processes due to unmet expectations.

**Relationship with central government.** Within the decentralised system, the central government [represented by the ministry of health, Finance and local government among others continued to play an oversight role related to sharing targets, budgets, finances and rules and procedures. The influence of these agencies on the local government level and the powerlessness of the local government actors was still very evident. For example, despite decentralization, the existence of many rules related to the planning process, dependence on the centre for conditional grants, pre-determined priorities with limited room for flexibility affected limited decision space of sub-national actors to plan and implement the district, facility, and community priorities. For example, some of the district respondents, as highlighted by the quote below,

expressed frustration at the top-down nature of engagements with the central government, with little room for flexibility or creativity, and consequences of challenging the status quo.

> "[…] Governments generally are a strait jacket. If you try to do creative, thinking that you are doing a noble cause, you will 'burn your fingers'". (District Respondent _11)

**Implementing partners: power, influence and complex relationship with district actors.** The presence and influence of the implementing partners at the subnational level cannot be underestimated across districts and seems even more required for districts that have demographic features that overwhelm the health system- such as refugees and being prone to epidemic outbreaks.

Despite having financial power and the related benefits that accrued to the districts, the presence of implementing partners at sub-national level was sometimes reported to come with challenges including stifling negotiation for district priorities, duplicative bureaucracies, and variations in planning cycles, and perceived lack of transparency with budgets, hence making the relationship between implementing partners and local government actors a complex one.

**Duplicative bureaucracies and related complexities.** The study found that many donors channeled resources to districts by sub-contracting implementing (Ips) to implement some of the programs at the subnational level instead of utilizing existing district structures.

> "We [IPs] do not give money directly to the districts […] when the activity is approved, we tell them [district team] to [go] ahead and start implementation. […] For example, for a meeting, we pay attendees via mobile [phone] money […] (District respondent_10)

> "But our[donor/IP] funding does not allow that [direct cash transfer to district accounts]. While we do budget support, it is in kind. We will not put money on the district account […] if its [is] the health worker who has done the activity, we send the money directly to that health worker […] (IP Respondent_19)

Overall, sub-contracting created a missed opportunity for supporting district capacity strengthening directly and for increasing the financial resources available locally to district leaders for implementing their work plans. Furthermore, subcontracting of implementing partners at subnational level was perceived to create duplicative bureaucracies which were perceived to expand the burden of reporting and accountability in the context of tight timelines. For example, multiple Implementing partners working at the sub-national level were reported to have introduced an extra layer of data, requisition and reporting structures, tools, and unharmonized guidelines for reporting and accountability in addition to those from central government.

When probed, IP respondents indicated that creating separate bureaucracies arose due to increasing concerns about local corruption, delays in accountability, and a greater emphasis on increasing value for money from donor agencies. These influenced IP practices such as not extending direct transfer of funds to districts or electronically wiring payments directly to beneficiaries or participants.

Additionally, respondents reported variations in planning cycles and priorities for IPs and districts, which created distortions and made harmonization with district priorities difficult.

> " […] Even before we have our budget approval for the next year […] often, the district is already finalizing their budget and since we have not yet finalized our budget, we cannot commit funds to the district before the donor approves. The district cycle starts in October […] our cycle starts in July. Sometimes you find that there is a certain lag in between" (IP respondent_19)

However, the district respondents noted that they continuously tolerated the situation for fear of "losing" their support to other districts.

**Perceived lack of transparency with budgets by IPs**

Furthermore, the general perception among district respondents was that IPs were not as transparent as expected, specifically about available budgets, although they were supposed to plan together, and this complicated planning for service delivery at subnational level.

> "[…] Most of them [IPs] do not disclose [their budget]. They will say we are going to do this, but you may not know about the budget, how much [money] they must spend." (District Respondent _33)

**Plausible solutions in relation to structural and agency challenges identified.** In response to the finance related challenges, the respondents identified broad recommendations about what central government, donors and implementing partners should do to address some of the challenges. These recommendations were checked and validated during the dissemination workshops. However, the research team was not able to conduct activities for prioritisation of recommendations.

In relation to limited decision space challenges, recommendations for the central government included expansion of decision space for local government to allow them to allocate resources according to community priorities

To address the issue of the creation of separate bureaucracies, participants recommended that donors should directly finance health systems directly rather than engaging third-party organizations, such as(implementing partners (Ips), which may absorb extensive financial and operational costs for HSS activities, resources that could be directly invested into district systems. In the short term, Governments and donors should consider pooling of funds to reduce waste as well as eliminating third parties' costs.

In relation to addressing and aligning with district priorities, participants recommended ball implementing partners focusing on systems strengthening at sub-national level should prioritize "hardware" investments, e.g., infrastructure, health workforce strengthening. "Software" investments, e.g., Data for planning, IT, community engagement for accountability will gain more traction after hardware gaps have been addressed.

To address the challenge of inadequate and delayed finances, Donors and central government need to incentivize practices that support health systems strengthening at sub-national level such as improving disbursement delays. Furthermore, the central government needs to increase discretionary funding for PHC to enable action on local health priorities.

Overlayered rules were a main challenge, and participants recommended that rules need to be simplified and that stakeholders should agree on a process to improve rational adoption of new rules and innovations, which should be subjected to an Impact Assessment to determine feasibility of implementation and scale-up prior to adoption.

Community participation: Community participation requirements should be relaxed to avoid generating mistrust over nonresponsive budgets. Stakeholders seeking to engage community members in planning processes should be clear and transparent about the definition, scope, and purpose of community participation

## Discussion

The performance of subnational/local governments in the health sector has remained sub-optimal despite consensus on the importance of subnational health systems to the delivery of quality, affordable, and equitable health services, as well as despite heavy investments in HSS. The findings from our application of the PEA framework accentuate the fact that to understand a problem, you must reflect on the underlying characteristics of the problem as well as its root causes [37]. In Uganda, we found that the problem of sub-optimal performance of the health system at sub-national level was driven by a constellation of factors that limited subnational decision space and created barriers to implementation. These include

several complex dilemmas: inadequate resources as well as underspent budgets, insufficient management capacity at the local level as well as duplicative bureaucracies, a proliferation of external actors engaged in Subnational government decision-making, often at the expense of engaging with local communities, and unrealized implementation plans and underutilized data, resulting in health plans and budgets inadequately responsive to community-generated priorities [30].

These health system challenges reflect, in part, an incomplete decentralization process in Uganda. While district health teams at the subnational level have taken on increasing responsibility for health service delivery, fiscal and planning autonomy are yet to be achieved. Subnational governments remain dependent on the central government for funding, which often oversteps their mandate and dictates subnational government plans through conditional grants. In comparison with Malawi [32] and Kenya [31], the situation of dependency on central government for funding was unique to Uganda [33]. While incomplete decentralization is common in other African settings, and has relevance in health, as well as other sector, our findings also underscore the importance of understanding this phenomenon in each context, considering the history of decentralization overall. Knowing where a country falls on the continuum of decentralization is needed in order to contextualize and appreciate the challenges related to the autonomy and the effective management of resources available to sub-national units [33]. Further reflections on the role of ongoing decentralization among the larger study's countries are summarized by Rodriguez et al [33].

The challenges identified further reflect the complex dynamics between a country's bureaucracy and external funders. In Uganda, subnational level actors are similarly beholden to donors and IPs that are active in their districts. Subnational actors rely on additional funding streams from external partners to supplement available financial resources. This influx of funding gives donors and IPs significant influence in defining district-level priorities. While donor funding is necessarily tied to a donor's organizational focus – and linked to their own processes – this often creates duplicative reporting streams that place additional administrative burden on DHMTs, and can skew prioritization. Streamlining bureaucratic processes and engagement to focus on shared priorities, including engagement at the national level and with donors and IPs, will be critical for enabling good governance at the subnational level.

Finally, inadequate financing emerged as the main factor hindering DHMTs from advancing HSS efforts and meeting the ever-growing population needs and demands. The proportion of the national budgetary allocation to health sector continues to fall short of the Abuja declaration recommended target of 15% [38,39], albeit a slight increment from 7.8% in 2019/20 [40]to 9.8% in FY 2021/22 [41]. This finding has implications for strategies to empower district governments with revenue as well as increased bargaining power to negotiate with funders to address local priorities.

## Study strengths and limitations

This study is the first problem-based political economy analysis of its kind, and the most recent one conducted in Uganda. Together with its sister studies in Malawi and Kenya, it richly updates the prior literature reflecting on decentralization, as well as on the consequences of this process on the health sector, in Uganda and more broadly in Eastern Africa.

Nevertheless, this analysis had several limitations. First, it was not possible to obtain all the necessary financial data to conduct a complete financial analysis as well as interpretation. The financial data needed for the analysis of available budgets for specific health service delivery and technical areas was largely unavailable and with many missing data points. Second, our study did not examine the role of the private not for profit sector, which plays an important role filling in gaps in public service delivery in Uganda. Given our focus on health planning and financing, we also were unable to explore all aspects of the political context in Uganda which may influence subnational operations. Third, due to COVID-19 pandemic restrictions and effects prevented the study team from capturing the effects on district planning fully and restricted engagement with national-level stakeholders through video interviews. Repeat interviews were not possible, so there was only limited follow-up to emerging themes. Saturation was not achieved in all themes and sub-themes, and a cross-case comparison among our case districts was not possible.

## Conclusion

In conclusion, the Political Economy Analysis lens is a useful approach to understanding why health systems strengthening interventions at subnational level in Uganda have stagnated despite massive investments. The challenges for HSS are embedded in the structural reality as well as agency interests, power-relations and actions, which are often at the expense of the local government's means and decision space. An extensive examination of this reality by the people that are signatories and control resource disbursement and donors will thus help to address the massive challenges highlighted in the analysis. The recommendations provided by respondents remain a wish list, because the district representatives are largely alienated from the centrally and donor driven priorities and decision making and further constrained by the related bureaucracies. Future research on health systems strengthening should not overlook the political economy dimensions present within subnational government units and its influence on resource allocation as well as the performance of implementing district work plans with existing resources. Despite the donor support being significant in health systems strengthening at subnational level, they need to be guided by local realities and priorities and should be flexible in their financing. The tension between national and subnational units will likely remain and will continue to be problematic until subnational units have a bigger say and control over the way the resources are allocated, capacity necessary to lead their programs in a truly decentralized fashion.

## Supporting information

**S1 Data. UNICEF PEA Framework Analysis Themes.** This supplementary file presents thematic data from a Political Economy Analysis (PEA) focused on health system dynamics across selected districts. The data is organized according to the UNICEF-adapted PEA framework and includes the following sheets: - READ ME: Lists stakeholder affiliations, governance levels, and participating districts or counties involved in the data collection and analysis. - OVERALL: Summarizes overarching themes and observations emerging from the entire analysis. - Structural-Context: Captures contextual and structural enablers and constraints (e.g., historical, socio-political, and economic factors) affecting health systems. - Structural-Rules: Details formal and informal rules, norms, and institutional arrangements shaping actor behavior and service delivery. - Agency Diagnosis: Presents themes related to actors' roles, interests, incentives, and relationships, highlighting how these influence decision-making and implementation. - Implementation+Pathways: Documents practical implementation challenges and pathways, including adaptive strategies and policy-practice gaps. - NOTES: Contains supplementary notes, possibly including researcher reflections or coding comments. Each sheet comprises synthesized qualitative insights coded from key informant interviews and other data sources, organized by district and thematic focus.
(DOCX)

## Acknowledgments

The COVID-19 pandemic broke out during our data collection period and increased the pressure and demands on national and local decision makers. Our appreciation goes to all the individuals who agreed to participate in this study at both national and subnational level. Special thanks to district level managers who worked tirelessly to ensure continued health service delivery for local communities during the Pandemic. We thank UNICEF-Eastern and Southern Africa Regional Office, UNICEF country office in Uganda and Ministry of Health in Uganda for their feedback and support in disseminating findings and validation of recommendations. We also acknowledge the commitment from the members of the study team at School of Public Health, Makerere University. These include the core team (Freddie Ssengooba, Justine Namakula, Xavier Nsabagasani) and research assistants- Adelaine Aryaija-Karemani, Rachael Bakubi, Paul Mubiri, Resty Nakayima, Tenywa Ronald, Robina Kyomuhendo and Sarah Elayeete. Without their effort, in the face of the COVID-19 pandemic, this study would not have been possible.

## Author contributions

**Conceptualization:** Justine Namakula, Xavier Nsabagasani, Ligia Paina, Abigail Neel, Daniela C. Rodriguez, Freddie Ssengooba.

**Data curation:** Justine Namakula, Xavier Nsabagasani, Ligia Paina, Daniela C. Rodriguez.

**Formal analysis:** Justine Namakula, Xavier Nsabagasani, Ligia Paina, Abigail Neel, Daniela C. Rodriguez, Freddie Ssengooba.

**Funding acquisition:** Ligia Paina, Daniela C. Rodriguez, Freddie Ssengooba.

**Investigation:** Justine Namakula, Xavier Nsabagasani, Freddie Ssengooba.

**Methodology:** Justine Namakula, Xavier Nsabagasani, Ligia Paina, Daniela C. Rodriguez, Freddie Ssengooba.

**Project administration:** Justine Namakula, Ligia Paina, Abigail Neel, Daniela C. Rodriguez, Freddie Ssengooba.

**Supervision:** Ligia Paina, Daniela C. Rodriguez, Freddie Ssengooba.

**Validation:** Justine Namakula, Xavier Nsabagasani, Chimwemwe Msukwa, Freddie Ssengooba.

**Visualization:** Freddie Ssengooba.

**Writing – original draft:** Justine Namakula, Xavier Nsabagasani, Freddie Ssengooba.

**Writing – review & editing:** Justine Namakula, Xavier Nsabagasani, Ligia Paina, Abigail Neel, Chimwemwe Msukwa, Daniela C. Rodriguez, Freddie Ssengooba.

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
