## [Decision Letter · Decision Letter 0]

PGPH-D-23-02177

What slows the progress of health systems strengthening at subnational level? A political Economy Analysis of three districts in Uganda

Dear Authors,

Thank you for submitting your manuscript to PLOS Global Public Health. After careful consideration, we feel that it has merit but does not fully meet PLOS Global Public Health’s publication criteria as it currently stands. Therefore, we invite you to submit a revised version of the manuscript that addresses the points raised during the review process.

We look forward to receiving your revised manuscript.

Kind regards,

Genevieve Cecilia Aryeetey, Ph.D

Academic Editor

Journal Requirements:

Additional Editor Comments (if provided):

Reviewers' comments:

Reviewer's Responses to Questions

**Comments to the Author**

1. Does this manuscript meet PLOS Global Public Health’s publication criteria?

Reviewer #1: Yes

Reviewer #2: Yes

Reviewer #3: Yes

2. Has the statistical analysis been performed appropriately and rigorously?

Reviewer #1: N/A

Reviewer #2: Yes

Reviewer #3: N/A

3. Have the authors made all data underlying the findings in their manuscript fully available (please refer to the Data Availability Statement at the start of the manuscript PDF file)?

Reviewer #1: Yes

Reviewer #2: No

Reviewer #3: Yes

4. Is the manuscript presented in an intelligible fashion and written in standard English?

Reviewer #1: Yes

Reviewer #2: Yes

Reviewer #3: Yes

Reviewer #1: Manuscript: What slows the progress of health systems strengthening at subnational level? A political Economy Analysis of three districts in Uganda

Review report

General comments

The manuscript examines performance of district health systems in Uganda, which is of great importance given continued failures to meet several health targets in different parts of the country and other African countries. The authors applied political economy analysis which is suitable to explore the issue at hand although, the PEA being a core part of the paper, is under-described. A number of issues (unique and interrelated) are reported to contribute to the poor performance of district health systems. The authors make several claims there in but some are inadequately backed by quotes from the interviews done. It seems that district health systems are primarily limited by economic forces/processes which are influenced by political processes.

The major comment relates to the discussion section of the manuscript, authors should add some context from other settings/studies especially for the two issues raised regarding decentralization and donor support.

Below are more specific comments for the authors to consider by section.

Introduction

The authors provide a good introduction of different elements of their study – UHC, HSS and district health system, highlighting the gap regarding HSS. However, it would be helpful if the authors also briefly introduced the PEA concept which seems to be a key point of emphasis in the manuscript.

Methods

1. The text related to table 1 (Pg 7, L168-169), could the authors clarify the “case selection criteria”. If it is the characteristics listed in the table, the authors should improve the statement to make that clear.

2. Pg8, L178 – ‘PEA’ has not been defined before. Authors should include the acronym at the first point where it was spelled out.

3. Pg 9, L195- The authors indicate that over 40 documents were reviewed, could the provide a list of these documents as an appendix, so that we know which documents contributed to the findings presented.

4. Between the introduction and methods, the authors should consider providing a little more detail on the PEA as applied to the present study.

5. Pg10 L213 – the authors should add some detail on the analysis/synthesis that led to the findings presented in the paper despite indicating that detailed steps are provided somewhere else, unless the purpose of the two papers was the same. Otherwise, some analysis details for the present study should be added.

Results

1. Pg 11, L240 – the opening statement, is it based on the findings of the study? Because it did not come out clearly in the prior sections. If it’s from findings, then it should be stated as such otherwise it might need to be referenced.

2. Financing

a. Sub-theme – insufficient resources …..Authors state that “KII suggested inadequate financial and non-financial resources…….” (Pg11, L253-254). However, the text largely focuses on financial resources and perhaps the sub-theme should be phrased as such (which is what was used in the summary under financing). Otherwise authors should add a description of the non-financial resources.

b. The quote under delayed disbursement Pg13, L301 to Pg14, L306; since the narrative seems to be inclined towards the three districts 1,2 and 3; perhaps the authors specify which of the three, respondent 7 is. Or the quote is moved to just after the description of districts 1 and 2,

c. Pg 15 L328-330, authors indicate there was consistency in availability of funds including maternal and child health. However, table 3 shows data were not available for most points for MCH. How did the authors reach the aforementioned deduction?

d. Pg15, L338 – starting with “for example….” Authors should revise the sentence for clarity

e. Pg 17, L378 – choose between attributed or blamed

f. Edit point 4 in table 4

3. Planning

a. The quotes to back-up claims under sub-theme one; Pg23 L512- the sentence is not clear

b. Pg23, L519-521- the first quote (resp 12) does not seem to really fit within the sub-theme in question (distorted accountability mechanisms)

c. Under the sub-theme of ‘variations in planning cycle – it would be good for the quote (from the district respondent) used to support the claim made by the authors in L545-546 (Pg24), which is not the case with the quote currently provided by the authors.

d. Table 5, bullet 2 seems like a solution for financing rather than planning

Discussion

1. Pg27,L607-608- could the authors clarify what situation they are referring to?

2. The discussion does not relate the study’s findings to what is happening elsewhere, is there a reason why this is so? There are two points the authors raise therein including failures in decentralization, this could be a good discussion point for other African settings or settings where decentralization has (or not) worked.

3. Pg28, L638- revise the statement for clarity (the third limitation)

Conclusion

1. Pg29, L654-655- Future research on what?

Reviewer #2: • Introduction

Overall, the introduction section is well-written.

Page 4, line 97: the definition of UHC should not be limited to health insurance

• Methods

o The authors should provide details on the data collection instruments (interview guides and data extraction sheet) including issues that were covered in each instrument

o The authors should provide a justification for the sample size that was reached. Was saturation achieved? At what point was saturation achieved?

o It is not clear to me why the document reviews were needed?

o Authors should include information on coding (type and number of people involved) in the data analysis section. Was inter-coder reliability achieved?

o How did the authors manage to manually analyzed 49 transcripts? Why did the authors not employ a qualitative data analysis assisted software

o The results are very detailed but could have benefited from a clear structure probably guided by an existing theory or analytical framework. As it stands, the results read more like a project report than that of a journal article.

o There are a number of theories and frameworks relating to political economy analysis. I will recommend that the authors should be guided by such frameworks in the presentation and discussion of the findings

Reviewer #3: Summary: This study was based on the topic “What slows the progress of health systems strengthening at subnational level? A political economy analysis of three districts in Uganda”. The subject is important and the authors should be applauded on their choice of topic.

There are a few major and some minor revisions present in the current submission that need to be addressed. Since this is a very important topic, I would like to strongly encourage the authors to consider addressing the issues raised to strengthen the work for consideration.

Kindly find it attached

**Do you want your identity to be public for this peer review?** For information about this choice, including consent withdrawal, please see our Privacy Policy

Reviewer #1: No

Reviewer #2: No

Reviewer #3: No

---

## [Decision Letter · Decision Letter 1]

PGPH-D-23-02177R1

What slows the progress of health systems strengthening at subnational level? A political Economy Analysis of three districts in Uganda

Dear Dr. Namakula,

Thank you for submitting your manuscript to PLOS Global Public Health. After careful consideration, we feel that it has merit but does not fully meet PLOS Global Public Health’s publication criteria as it currently stands. Therefore, we invite you to submit a revised version of the manuscript that addresses the points raised during the review process.

In particular, we note that the previous comments and concerns raised by Reviewers 1 and 2 do not appear to have been addressed. The response to the reviewer comments provided only refers to the comments provided by Reviewer 3. This issue has been raised by Reviewer 2 in their re-review, comments below. Please carefully reconsider all of the previous comments provided by both Reviewer 1 and Reviewer 2 and revise the manuscript accordingly.

We look forward to receiving your revised manuscript.

Kind regards,

Helen Howard

Staff Editor

Journal Requirements:

Additional Editor Comments (if provided):

Reviewers' comments:

Reviewer's Responses to Questions

**Comments to the Author**

Reviewer #2: (No Response)

Reviewer #3: All comments have been addressed

publication criteria?

Reviewer #2: Yes

Reviewer #3: Yes

3. Has the statistical analysis been performed appropriately and rigorously?

Reviewer #2: N/A

Reviewer #3: N/A

4. Have the authors made all data underlying the findings in their manuscript fully available (please refer to the Data Availability Statement at the start of the manuscript PDF file)?

Reviewer #2: Yes

Reviewer #3: Yes

5. Is the manuscript presented in an intelligible fashion and written in standard English?

Reviewer #2: No

Reviewer #3: Yes

Reviewer #2: There is no evidence that the authors addressed the comments from my first round of review. I cannot find in the response to authors document how the authors have addressed my comments. I repeat the comments below hoping that the authors will respond to them this time round.

• Introduction

Overall, the introduction section is well-written.

Page 4, line 97: the definition of UHC should not be limited to health insurance

• Methods

o The authors should provide details on the data collection instruments (interview guides and data extraction sheet) including issues that were covered in each instrument

o The authors should provide a justification for the sample size that was reached. Was saturation achieved? At what point was saturation achieved?

o It is not clear to me why the document reviews were needed?

o Authors should include information on coding (type and number of people involved) in the data analysis section. Was inter-coder reliability achieved?

o How did the authors manage to manually analyzed 49 transcripts? Why did the authors not employ a qualitative data analysis assisted software

o The results are very detailed but could have benefited from a clear structure probably guided by an existing theory or analytical framework. As it stands, the results read more like a project report than that of a journal article.

o There are a number of theories and frameworks relating to political economy analysis. I will recommend that the authors should be guided by such frameworks in the presentation and discussion of the findings

Reviewer #3: I have reviewed the revised manuscript and think all the concerns I raised in the previous version have been satisfactorily addressed. Congratulations to the authors for the excellent work, which contributes to our understanding of health systems strengthening at the sub-national level through a political economy analysis lens.

**Do you want your identity to be public for this peer review?** For information about this choice, including consent withdrawal, please see our Privacy Policy

Reviewer #2: No

Reviewer #3: **Yes: ** Dominic Dormenyo Gadeka, MPH, PhD

******The previous review by Reviewer 1 is copied below:**

Reviewer #1: Manuscript: What slows the progress of health systems strengthening at subnational level? A political Economy Analysis of three districts in Uganda

Review report

General comments

The manuscript examines performance of district health systems in Uganda, which is of great importance given continued failures to meet several health targets in different parts of the country and other African countries. The authors applied political economy analysis which is suitable to explore the issue at hand although, the PEA being a core part of the paper, is under-described. A number of issues (unique and interrelated) are reported to contribute to the poor performance of district health systems. The authors make several claims there in but some are inadequately backed by quotes from the interviews done. It seems that district health systems are primarily limited by economic forces/processes which are influenced by political processes.

The major comment relates to the discussion section of the manuscript, authors should add some context from other settings/studies especially for the two issues raised regarding decentralization and donor support.

Below are more specific comments for the authors to consider by section.

Introduction

The authors provide a good introduction of different elements of their study – UHC, HSS and district health system, highlighting the gap regarding HSS. However, it would be helpful if the authors also briefly introduced the PEA concept which seems to be a key point of emphasis in the manuscript.

Methods

1. The text related to table 1 (Pg 7, L168-169), could the authors clarify the “case selection criteria”. If it is the characteristics listed in the table, the authors should improve the statement to make that clear.

2. Pg8, L178 – ‘PEA’ has not been defined before. Authors should include the acronym at the first point where it was spelled out.

3. Pg 9, L195- The authors indicate that over 40 documents were reviewed, could the provide a list of these documents as an appendix, so that we know which documents contributed to the findings presented.

4. Between the introduction and methods, the authors should consider providing a little more detail on the PEA as applied to the present study.

5. Pg10 L213 – the authors should add some detail on the analysis/synthesis that led to the findings presented in the paper despite indicating that detailed steps are provided somewhere else, unless the purpose of the two papers was the same. Otherwise, some analysis details for the present study should be added.

Results

1. Pg 11, L240 – the opening statement, is it based on the findings of the study? Because it did not come out clearly in the prior sections. If it’s from findings, then it should be stated as such otherwise it might need to be referenced.

2. Financing

a. Sub-theme – insufficient resources …..Authors state that “KII suggested inadequate financial and non-financial resources…….” (Pg11, L253-254). However, the text largely focuses on financial resources and perhaps the sub-theme should be phrased as such (which is what was used in the summary under financing). Otherwise authors should add a description of the non-financial resources.

b. The quote under delayed disbursement Pg13, L301 to Pg14, L306; since the narrative seems to be inclined towards the three districts 1,2 and 3; perhaps the authors specify which of the three, respondent 7 is. Or the quote is moved to just after the description of districts 1 and 2,

c. Pg 15 L328-330, authors indicate there was consistency in availability of funds including maternal and child health. However, table 3 shows data were not available for most points for MCH. How did the authors reach the aforementioned deduction?

d. Pg15, L338 – starting with “for example….” Authors should revise the sentence for clarity

e. Pg 17, L378 – choose between attributed or blamed

f. Edit point 4 in table 4

3. Planning

a. The quotes to back-up claims under sub-theme one; Pg23 L512- the sentence is not clear

b. Pg23, L519-521- the first quote (resp 12) does not seem to really fit within the sub-theme in question (distorted accountability mechanisms)

c. Under the sub-theme of ‘variations in planning cycle – it would be good for the quote (from the district respondent) used to support the claim made by the authors in L545-546 (Pg24), which is not the case with the quote currently provided by the authors.

d. Table 5, bullet 2 seems like a solution for financing rather than planning

Discussion

1. Pg27,L607-608- could the authors clarify what situation they are referring to?

2. The discussion does not relate the study’s findings to what is happening elsewhere, is there a reason why this is so? There are two points the authors raise therein including failures in decentralization, this could be a good discussion point for other African settings or settings where decentralization has (or not) worked.

3. Pg28, L638- revise the statement for clarity (the third limitation)

Conclusion

1. Pg29, L654-655- Future research on what?

---

## [Decision Letter · Decision Letter 2]

PGPH-D-23-02177R2

What slows the progress of health systems strengthening at subnational level? A political Economy Analysis of three districts in Uganda

Dear Dr. Namakula,

Thank you for submitting your manuscript to PLOS Global Public Health. After careful consideration, we feel that it has merit but does not fully meet PLOS Global Public Health’s publication criteria as it currently stands. Therefore, we invite you to submit a revised version of the manuscript that addresses the points raised during the review process.

Unfortunately, the two reviewers who provided comments on the previous versions of your manuscript were not available, so we have invited a new reviewer who has provided some valuable feedback. Please see the reviewer's comments below. 

Could you please revise the manuscript to carefully address the concerns raised?

We look forward to receiving your revised manuscript.

Kind regards,

Steve Zimmerman, PhD

PLOS Staff Editor

Additional Editor Comments (if provided):

Reviewers' comments:

Reviewer's Responses to Questions

**Comments to the Author**

Reviewer #4: (No Response)

publication criteria?

Reviewer #4: Partly

3. Has the statistical analysis been performed appropriately and rigorously?

Reviewer #4: N/A

4. Have the authors made all data underlying the findings in their manuscript fully available (please refer to the Data Availability Statement at the start of the manuscript PDF file)?

Reviewer #4: Yes

5. Is the manuscript presented in an intelligible fashion and written in standard English?

Reviewer #4: Yes

Reviewer #4: The manuscript addresses an important theme of strengthening health systems in Uganda. Several areas may need more attention from the authors:

1. The title indicates that the paper will provide a political economy analysis in Uganda to understand “why health systems are failing to improve despite marked investments.” However, the paper does not include a political analysis to provide a clear picture of the political context in Uganda in which health systems are evolving and development assistance is provided. As it currently reads, the paper is mainly focusing on two building blocks of health systems: governance and financing. It may be better to pursue this approach and include a problem statement in the introduction section so the reader can follow it more easily.

2. The paper repeatedly refers to poor-performing health systems. However, no statistical evidence is provided in terms of health outcomes to support this claim.

3. The paper needs more literature review about the donor-government relationships and development assistance for health in general as the issues identified do not seem new (e.g. need for the general budget support, the mismatch between the donor priorities and country needs, etc.)

4. The quotes were not very helpful. Perhaps the authors can find better quotes to support their arguments.

5. There is a lot of negativity toward donors and implementing partners, which needs to be addressed strategically at the national level. It seems naïve to assume that aid comes with no strings attached.

**Do you want your identity to be public for this peer review?** For information about this choice, including consent withdrawal, please see our Privacy Policy

Reviewer #4: No

---

## [Editor Report · Decision Letter 3]

What slows the progress of health systems strengthening at subnational level? A political Economy Analysis of three districts in Uganda

PGPH-D-23-02177R3

Dear Dr. Namakula,

We are pleased to inform you that your manuscript 'What slows the progress of health systems strengthening at subnational level? A political Economy Analysis of three districts in Uganda' has been provisionally accepted for publication in PLOS Global Public Health.

Best regards,

Julia Robinson

Executive Editor